# Semi-Supervised Contrastive Learning with Orthonormal Prototypes

## Abstract

Contrastive learning has emerged as a powerful method in deep learning, excelling at learning effective representations through contrasting samples from different distributions. However, dimensional collapse, where embeddings converge into a lower-dimensional space, poses a significant challenge, especially in semi-supervised and self-supervised setups. In this paper, we propose CLOP, a novel semi-supervised loss function designed to prevent dimensional collapse by promoting the formation of orthogonal linear subspaces among class embeddings. Through extensive experiments on real and synthetic datasets, we demonstrate that CLOP improves performance in image classification and object detection tasks while also exhibiting greater stability across different learning rates and batch sizes.

## Introduction

Recent advancements in deep learning have positioned Contrastive Learning as a leading paradigm, largely due to its effectiveness in learning representations by contrasting samples from different distributions while aligning those from the same distribution. Prominent models in this domain include SimCLR Chen et al. (2020), Contrastive Multiview Coding (CMC) Tian et al. (2020a), VICReg Bardes et al. (2021), BarLowTwins Zbontar et al. (2021), among others Henaff (2020); Li et al. (2020); Wu et al. (2018). These models share a common two-stage framework: representation learning and fine-tuning. In the first stage, representation learning is performed in a self-supervised manner, where the model is trained to map inputs to embeddings using contrastive loss to separate samples from different labels. In the second stage, fine-tuning occurs under a supervised setup, where labeled data is used to classify embeddings correctly. For practical applicability, a small amount of labeled data is required in the fine-tuning stage to produce meaningful classifications, making the overall pipeline semi-supervised. Empirical evidence demonstrates that these models, even with limited labeled data (as low as 10%), can achieve performance comparable to fully-supervised approaches on moderate to large datasets Jaiswal et al. (2020).

Despite the effectiveness of contrastive learning on largely unlabeled datasets, a common issue encountered during the training process is dimensional collapse. As pointed out by Fu et al. (2022); Gill et al. (2024); Hassanpour et al. (2024); Jing et al. (2021); Rusak et al. (2022); Tao et al. (2024); Xue et al. (2023), this phenomenon describes the collapse of output embeddings from the neural network into a lower-dimensional space, reducing their spatial utility and leading to indistinguishable classes. There are two main approaches to resolve this issue: augmentation modification Fu et al. (2022); Jing et al. (2021); Tao et al. (2024); Xue et al. (2023) and loss modification Fu et al. (2022); Hassanpour et al. (2024); Rusak et al. (2022). In this paper, our approach selects prototypes similarly to Gill et al. (2024); Zhu et al. (2022). The key distinction is that our method aims to push the embeddings toward distinct orthogonal linear subspaces, allowing them to occupy a higher-rank space. We demonstrate through experiments that CLOP is more effective for image classification and object detection tasks.

The main contribution of this paper is to address the issue of dimensional collapse in contrastive learning losses that rely solely on cosine similarity. We first show that such losses admit degenerate stationary points where embeddings collapse into a rank-1 subspace, rendering the learned representations uninformative. To overcome this limitation, we introduce CLOP, a novel contrastive loss that promotes clustering of embeddings around a set of orthonormal prototypes, thereby preserving representational diversity. CLOP naturally extends to semi-supervised and fully-supervised settings,

making it particularly effective when only a subset of the training data is labeled. Through extensive experiments on embedding visualization, image classification, and object detection across balanced and imbalanced datasets, we demonstrate that CLOP consistently outperforms baseline methods, exhibiting superior robustness and stability under varying learning rates and reduced batch sizes.

**Paper Organization** We first provide essential background information and discuss recent advancements in both self-supervised and supervised contrastive learning, as well as analyzing the dimensional collapse phenomenon associated with contrastive learning methods. Then, we elaborate on the motivation behind our approach through simulations and detailed gradient analysis. Subsequently, we introduce our proposed model, CLOP. Finally, we present extensive experimental evaluations conducted on image datasets.

## RELATED WORK

Contrastive learning has gained prominence in deep learning for its ability to learn meaningful representations by pulling together similar (positive) pairs and pushing apart dissimilar (negative) pairs in the embedding space. Positive pairs are generated through techniques like data augmentation, while negative pairs come from unrelated samples, making contrastive learning particularly effective in self-supervised tasks like image classification. Pioneering models such as SimCLR Chen et al. (2020), CMC Tian et al. (2020a), VICReg Bardes et al. (2021), and Barlow Twins Zbontar et al. (2021) share the objective of minimizing distances between augmented versions of the same input (positive pairs) and maximizing distances between unrelated inputs (negative pairs). SimCLR maximizes agreement between augmentations using contrastive loss, while CMC extends this to multi-view learning Chen et al. (2020); Tian et al. (2020a). VICReg introduces variance-invariance-covariance regularization without relying on negative samples Bardes et al. (2021), and Barlow Twins reduce redundancy between different augmentations Zbontar et al. (2021).

Recent innovations have improved contrastive learning across various domains. For instance, methods like structure-preserving quality enhancement in CBCT images Kang et al. (2023) and false negative cancellation Huynh et al. (2022) have enhanced image quality and classification accuracy. In video representation, cross-video cycle-consistency and inter-intra contrastive frameworks Wu & Wang (2021); Tao et al. (2022) have shown significant gains. Additionally, contrastive learning has advanced sentiment analysis Xu & Wang (2023), recommendation systems Yang et al. (2022a), and molecular learning with faulty negative mitigation Wang et al. (2022b). Xiao et al. (2024) introduces GraphACL, a novel framework for contrastive learning on graphs that captures both homophilic and heterophilic structures without relying on augmentations.

## CONTRASTIVE LOSS

In unsupervised learning, Wu et al. (2018) introduced InfoNCE, a loss function defined as:

$$\mathcal{L}_{\text{infoNCE}} = -\sum_{i \in I} \log \frac{\exp(\mathbf{z}_i^\top \mathbf{z}_{j(i)}/\tau)}{\sum_{a \neq i} \exp(\mathbf{z}_i^\top \mathbf{z}_a/\tau)} \tag{1}$$

where $\mathbf{z}_i$ is the embedding of sample $i$, $j(i)$ its positive pair, and $\tau$ controls the temperature.

Recent refinements focus on (1) component modifications, (2) similarity adjustments, and (3) novel approaches. Li et al. (2020) use EM with k-means to update centroids and reduce mutual information loss, while Wang et al. (2022a) add L2 distance to InfoNCE, though both underperform state-of-the-art (SOTA) techniques. Xiao et al. (2020) reduce noise with augmentations, and Yeh et al. (2022) improve gradient efficiency with Decoupled Contrastive Learning, though neither surpasses SOTA. In similarity adjustments, Chuang et al. (2020) propose a debiased loss, and Ge et al. (2023) use hyperbolic embeddings, but neither outperforms SOTA. Novel methods include min-max InfoNCE Tian et al. (2020b), Euclidean-based losses Bardes et al. (2021), and dimension-wise cosine similarity Zbontar et al. (2021), achieving competitive performance without softmax-crossentropy.

## SEMI-SUPERVISED CONTRASTIVE LEARNING

Semi-supervised contrastive learning effectively leverages both labeled and unlabeled data to learn meaningful representations. Zhang et al. (2022) introduced a framework with similarity co-calibration

to mitigate noisy labels by adjusting the similarity between pairs. Inoue & Goto (2020) proposed Generalized Contrastive Loss (GCL), which unifies supervised and unsupervised learning for speaker recognition, while Kim et al. (2021) combined contrastive self-supervision with consistency regularization in SelfMatch. Sohn et al. (2020) introduced FixMatch, which combines pseudo-labeling with consistency regularization. In this approach, weakly augmented samples generate pseudo-labels that guide strongly augmented versions, ensuring robust semi-supervised learning (SSL) performance. Yang et al. (2022b) enhanced SSCL by enforcing class-wise consistency in learned representations, improving robustness to class imbalance and increasing generalization across datasets. Zheng et al. (2022) proposed SimMatch, a framework that unifies contrastive learning and consistency regularization by optimizing both instance-level alignment and class-level semantic consistency, leading to improved SSL feature representations. Building on this, Zheng et al. (2023) introduced SimMatch-V2, refining the balance between contrastive and consistency learning objectives, further enhancing transferability and performance in semi-supervised settings.

## DIMENSIONAL COLLAPSE

For dimensional collapse in contrastive learning, Jing et al. (2021) examine dimensional collapse in self-supervised learning. They attribute this to strong augmentations distorting features and implicit regularization driving weights toward low-rank solutions. Similarly, Xue et al. (2023) explore how simplicity bias leads to class collapse and feature suppression, with models favoring simpler patterns over complex ones. They suggest increasing embedding dimensionality and designing augmentation techniques that preserve class-relevant features to counter this bias and promote diverse feature learning. Fu et al. (2022) emphasize the role of data augmentation and loss design in preventing class collapse, proposing a class-conditional InfoNCE loss term that uniformly pulls apart individual points within the same class to enhance class separation. In supervised contrastive learning, Gill et al. (2024) propose loss function modifications to follow an ETF geometry by selecting prototypes that form this structure. In graph contrastive learning, Tao et al. (2024) introduce a whitening transformation to decorrelate feature dimensions, avoiding collapse and enhancing representation capacity. Finally, Rusak et al. (2022) investigate the preference of contrastive learning for content over style features, leading to collapse. They propose to leverage adaptive temperature factors in the loss function to improve feature representation quality.

## MOTIVATION

In this section, we analyze the phenomenon of dimensional collapse in contrastive learning. We first demonstrate that dimensional collapse represents a local stationary point of the InfoNCE loss by showing that linear embeddings result in a zero gradient (Lemma 1). Subsequently, we discuss how repulsive force within InfoNCE could induce a shift in the embedding mean through comparisons between pairs of negative samples, ultimately leading to dimensional collapse. While our demonstration utilizes the InfoNCE loss, this conclusion generalizes to most current loss functions relying solely on cosine similarity as their metric, encompassing unsupervised Henaff (2020); Chen et al. (2020); Cui et al. (2021); Xiao et al. (2020); Yeh et al. (2022); Wang et al. (2022a); Li & Pimentel-Alarcón (2024), semi-supervised Hu et al. (2021); Shen et al. (2021), and supervised contrastive learning Khosla et al. (2020); Cui et al. (2021); Peeters & Bizer (2022); Li et al. (2022).

The InfoNCE loss (Eq. (1)) aims to encourage the embeddings to form distinguishable clusters in high-dimensional space, thereby facilitating classification for downstream models. However, in Lemma 1, we demonstrate that the worst-case scenario — where all embeddings become identical or co-linear — also constitutes a local optimum for the InfoNCE loss. This observation suggests that, from a theoretical perspective, InfoNCE exhibits instability, as both the best and worst solutions can lead to stationary points.

**Lemma 1.** *Let $\mathcal{F} : \mathbb{R}^m \to \mathbb{R}^{m'}$ be a family of Contrastive Learning structures, where $m$ and $m'$ denote the dimensions of the inputs and embeddings, respectively. If a function $f \in \mathcal{F}$ is trained using the InfoNCE loss, then there exist infinitely many local stationary points where all embeddings produced by $f$ are all equal or co-linear.*

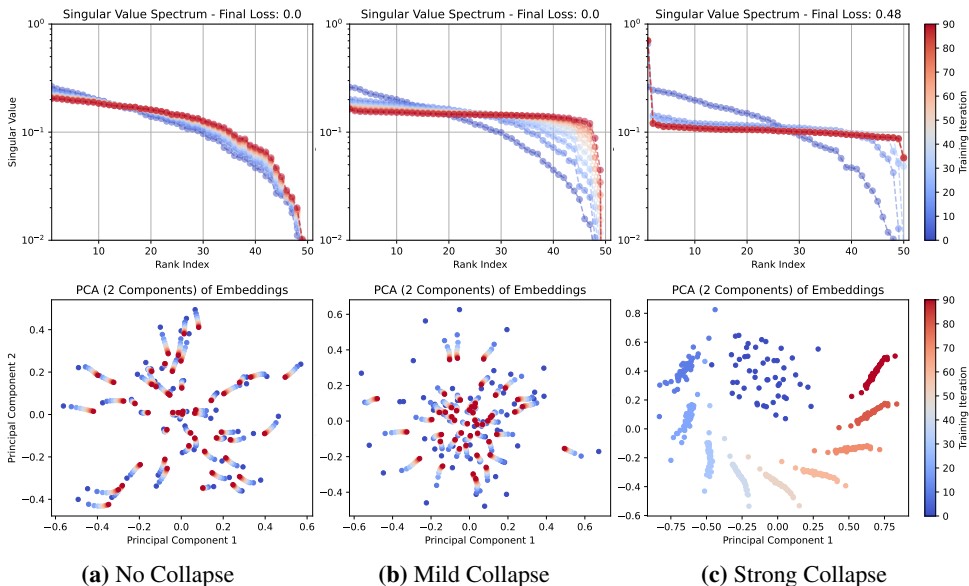

**(a)** No Collapse                **(b)** Mild Collapse                **(c)** Strong Collapse

Figure 1: Simulation with Repulsive Force on 50 simulated points in 50-dimensional space.

The proof of Lemma 1 relies on the observation that the embeddings are only compared against each other. If all embeddings are either identical or co-linear, the gradient vanishes due to the lack of angular differences, as well as the normalization process. The full proof is presented in Appendix.

To understand the dynamics of contrastive learning, it is crucial to consider two forces acting on each embedding: the *gravitational force* within the same pseudo class and the *repulsive force* between different pseudo classes. Contrary to the common belief that the gravitational force is responsible for inducing collapse, we observe that the overshooting of the repulsive force could be directly related to the dimensional collapse in contrastive learning.

To better illustrate the repulsive force, we conducted a simulation using 50 randomly generated embeddings in 50-d space, where positive pairs initially coincide at a single point, reflecting a scenario where the model has successfully merged augmented variants from the same input source into one embedding. We subsequently performed gradient descent on these embeddings using the InfoNCE loss with temperature of 0.1, recording the embedding trajectories throughout training. The simulations show three observations on No Collapse, Mild Collapse, and Strong Collapse, which are achieved using learning rates of 0.01, 0.1, and 1. The embedding singular value spectrum and the first two principal components of these embeddings are visualized in Figure 1. At Figure 1 (a), the repulsive force inherent to the InfoNCE loss effectively redistributes the embeddings across a more uniform space, as indicated by the more evenly dispersed singular values across the embedding dimensions. Conversely, at Figure 1 (c), the embeddings fail to redistribute and instead collapse into a one-dimensional subspace, corroborating the stationary point described by Lemma 1. Furthermore, analysis of the first two principal components clearly illustrates a significant shift in the embedding mean and a reduction in variance throughout the training iterations. This observation indicates that repulsive force could induce a substantial shift in the embedding mean, consequently accelerating the collapse process by aligning gradients in similar directions.

In the Appendix, we conducted a theoretical analysis of the gradient descent process using the InfoNCE loss. Briefly, our analysis reveals that, after performing a single gradient descent step, the upper bound on the norm of the embedding mean is scaled by the factor, involving minimum embedding norm before normalization and number of negative samples within the batch.

## CLOP: Populating Embedding Rank with Orthonormal Prototypes

To avoid the issue of embedding collapsing into a rank-1 linear subspace, we introduce a novel approach that promotes point isolation by adding an additional term to the loss function for contrastive

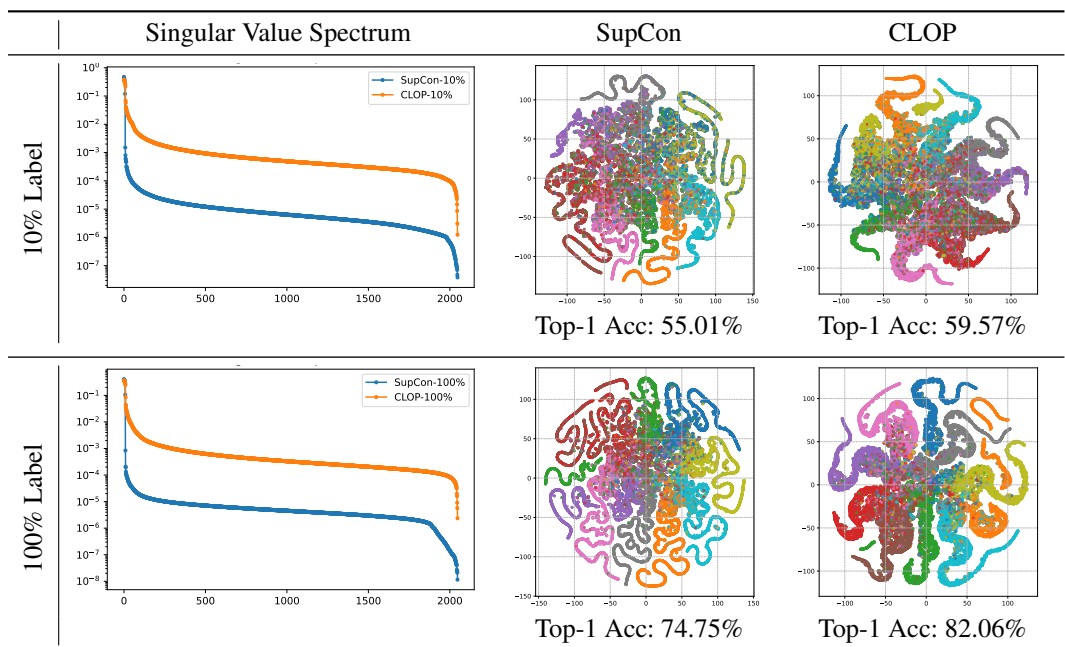

Table 1: SupCon vs. CLOP under different label usage

learning. Specifically, we initialize a group of *orthonormal prototypes*. The number of orthonormal prototypes matches the total number of classes in the dataset. We then maximize the similarity between the orthonormal prototypes and the labeled samples in the training set.

Formally, let $\mathcal{S}$ be the labeled training set containing pairs of embeddings and labels, denoted as $\mathcal{S} = \{(\mathbf{z}_i, y_i) \mid i \in \{1, \ldots, |\mathcal{S}|\}\}$. The set of prototypes, denoted as $\mathcal{C}$, is defined as $\mathcal{C} = \{\mathbf{c}_1, \ldots, \mathbf{c}_k\}$, where $k$ represents the number of classes in the dataset. To generate the prototypes $\mathcal{C}$, we randomly sample $k$ i.i.d. vectors from an $m'$-dimensional space, where $|\mathbf{z}_i| = m'$. Subsequently, we apply singular value decomposition (SVD) to obtain the orthonormal basis, denoted as $\mathcal{C}$. This ensures that each prototype $\mathbf{c}_i$ is initialized as a unit vector, orthogonal to all other prototypes, at the beginning of the training process. The CLOP loss is formulated as follows:

$$\mathcal{L}_{\text{CLOP}} = \mathcal{L}_{\text{infoNCE}} + \lambda \frac{1}{|\mathcal{S}|} \sum_{i=1}^{|\mathcal{S}|} (1 - s(\mathbf{z}_i, \mathbf{c}_{y_i})), \qquad (2)$$

where $s(\cdot, \cdot)$ denotes the similarity metric, typically chosen to be the same as that used in $\mathcal{L}_{\text{infoNCE}}$, namely cosine similarity.

The primary objective of the CLOP loss is to align all embeddings corresponding to the same class towards a common target prototype, $\mathbf{c}_{y_i}$. Beyond the "gravitational force" and "repulsive force" provided by the main contrastive loss, the CLOP loss introduces a supervised "pulling force" that prevents collapse by pulling labeled embeddings into class-specific orthogonal subspaces. It is important to note that, without additional constraints, samples outside of set $\mathcal{S}$ may still converge to other unspecified embeddings, potentially collapsing into a rank-1 subspace. However, a fundamental assumption in contrastive learning is that augmented samples are treated as being drawn from the same distribution as the original input data from the same class. Thus, the "gravitational force" between embeddings of the same class should pull unsupervised embeddings toward the target prototypes.

To assess the supervisory efficacy of CLOP, we visualized the CIFAR100's representations learned by both CLOP and SupCon under two label regimes (10% and 100%) in Table 1. Embeddings were projected to two dimensions using t-distributed Stochastic Neighbor Embedding (t-SNE) Van der Maaten & Hinton (2008), and we also examined each model's singular value spectrum. Both methods employ a ResNet-50 backbone with linear projection heads four times wider than standard (denoted as *ResNet-50 (4x)*), trained for 500 epochs with a batch size of 256 and an initial learning rate

| Methods | 10% Labels | | 50% Labels | |
|---|---|---|---|---|
| | Top-1 | Top-5 | Top-1 | Top-5 |
| SupCon | 0.595 | 0.865 | 0.625 | 0.884 |
| FixMatch | 0.588 | 0.883 | 0.611 | 0.887 |
| SsCL | 0.715 | 0.829 | 0.746 | 0.887 |
| CCSSL | 0.735 | 0.841 | 0.757 | 0.894 |
| SimMatch | 0.719 | 0.829 | 0.724 | 0.866 |
| SimMatch-V2 | 0.729 | 0.840 | 0.729 | 0.877 |
| CLOP | **0.743** | **0.904** | **0.76** | **0.92** |

Table 2: Top 1 and 5 accuracy on CIFAR-100.

| Methods | 10% Labels | | 50% Labels | |
|---|---|---|---|---|
| | Top-1 | Top-5 | Top-1 | Top-5 |
| SupCon | 0.704 | 0.874 | 0.734 | 0.830 |
| FixMatch | 0.720 | 0.886 | 0.774 | 0.911 |
| SsCL | 0.721 | 0.909 | 0.786 | 0.899 |
| CCSSL | 0.751 | 0.923 | 0.771 | 0.907 |
| SimMatch | 0.740 | **0.930** | 0.776 | 0.904 |
| SimMatch-V2 | 0.748 | 0.917 | 0.788 | 0.916 |
| CLOP | **0.791** | 0.927 | **0.829** | **0.949** |

Table 3: Top 1 and 5 accuracy on ImageNet.

| Methods | CIFAR-100 | | ImageNet-200 | | ImageNet | |
|---|---|---|---|---|---|---|
| | Top-1 | Top-5 | Top-1 | Top-5 | Top-1 | Top-5 |
| SupCon | 0.585 | 0.858 | 0.505 | 0.749 | 0.672 | 0.780 |
| FixMatch | 0.576 | 0.840 | 0.621 | 0.759 | 0.708 | 0.860 |
| SsCL | 0.719 | 0.860 | 0.585 | 0.733 | 0.730 | 0.862 |
| CCSSL | 0.744 | 0.874 | 0.631 | 0.889 | 0.710 | 0.865 |
| SimMatch | 0.685 | 0.829 | 0.527 | 0.768 | 0.705 | 0.855 |
| SimMatchV2 | 0.689 | 0.862 | 0.659 | 0.846 | 0.733 | 0.845 |
| CLOP | **0.763** | **0.918** | **0.689** | **0.898** | **0.799** | **0.932** |

Table 4: Top-1 and Top-5 accuracy on CIFAR-100, ImageNet-200, and ImageNet under imbalanced-class training.

of 1.0. CLOP yields a higher effective rank in its embedding space and produces more compact, well-separated clusters under both limited (10%) and full (100%) supervision. Quantitatively, linear fine-tuning on ImageNet confirms this advantage: at 10% label usage, CLOP achieves a top-1 accuracy of 59.57%, outperforming SupCon's 55.01% by 4.56%; at 100% label usage, CLOP reaches 82.06% versus SupCon's 74.75%, a gain of 7.31%. This experiment is conducted on a single NVIDIA A100 GPU, with each run completing within 4 hours of training.

## EXPERIMENT

In this section, we evaluate the performance of CLOP in various learning settings. First, we compare CLOP against existing semi-supervised contrastive learning methods on image classification tasks. Furthermore, we show that CLOP consistently outperforms competitors under both balanced and imbalanced class distributions. Next, we highlight the generalizability of CLOP by presenting its outstanding transfer learning results on image classification and object detection tasks. Finally, we conduct extensive ablation studies on key hyperparameters, including learning rate, batch size, $\lambda$, similarity metrics, and augmentation strategies, highlighting CLOP's robustness to varying learning rates and its effectiveness in small-batch learning.

### SEMI-SUPERVISED IMAGE CLASSIFICATION

For balanced-class training, we utilize the full CIFAR-100 Krizhevsky et al. (2009) and ImageNet Deng et al. (2009) datasets, considering scenarios where either $10\%$ or $50\%$ of labels are available for contrastive learning. CLOP is implemented with a ResNet-50 (4x) backbone using the SimCLR loss function. We benchmark CLOP against several state-of-the-art semi-supervised contrastive learning methods, including SupCon Khosla et al. (2020), FixMatch Sohn et al. (2020), SsCL Zhang et al. (2022), CCSSL Yang et al. (2022b), SimMatch Zheng et al. (2022), and SimMatch-V2 Zheng et al. (2023). The classification results for CIFAR-100 and ImageNet are presented in Table 2 and Table 3, respectively. Our findings indicate that CLOP consistently outperforms all competing methods across all experimental settings. Notably, on CIFAR-100, CLOP achieves the

| Method | Food | CIFAR10 | CIFAR100 | SUN397 | DTD | Caltech-101 | Flowers |
|---|---|---|---|---|---|---|---|
| SimCLR | 0.8820 | 0.9770 | 0.8590 | **0.6350** | 0.7320 | 0.9210 | 0.9700 |
| SupCon | 0.8723 | 0.9742 | 0.8427 | 0.5804 | 0.7460 | 0.9104 | 0.9600 |
| FixMatch | 0.8824 | 0.9639 | 0.8553 | 0.5774 | 0.7269 | 0.9123 | 0.9669 |
| SsCL | 0.8546 | 0.9866 | 0.8481 | 0.5800 | 0.7300 | 0.9115 | 0.9574 |
| CCSSL | 0.8663 | 0.9637 | 0.8352 | 0.5818 | 0.7270 | 0.9029 | 0.9456 |
| SimMatch | **0.8881** | 0.9759 | 0.8435 | 0.6004 | 0.7306 | 0.9013 | 0.9646 |
| SimMatch-V2 | 0.8568 | 0.9638 | 0.8270 | 0.5886 | **0.7526** | 0.9185 | 0.9613 |
| CLOP (this paper) | 0.8792 | **0.9989** | **0.8809** | 0.6267 | 0.7385 | **0.9331** | **0.9718** |

Table 5: Transfer learning results for classification tasks (pretrained on ImageNet). Numbers are mean-per-class accuracy for Caltech and Flowers; and top-1 accuracy for all other datasets.

| Method | Birdsnap | Cars | Aircraft | VOC2007 | Pets |
|---|---|---|---|---|---|
| SimCLR | 0.7590 | 0.9130 | 0.8785 | 0.8410 | 0.8920 |
| SupCon | 0.7515 | 0.9169 | 0.8409 | 0.8517 | 0.9347 |
| FixMatch | 0.7545 | 0.9004 | 0.8462 | 0.8372 | **0.9515** |
| SsCL | 0.7343 | 0.9089 | 0.8341 | 0.8504 | 0.9288 |
| CCSSL | 0.7613 | **0.9247** | 0.8528 | 0.8580 | 0.9471 |
| SimMatch | 0.7562 | 0.9127 | 0.8269 | 0.8393 | 0.9200 |
| SimMatchV2 | 0.7516 | 0.9186 | 0.8453 | 0.8546 | 0.9494 |
| CLOP | **0.7794** | 0.9171 | **0.8810** | **0.8646** | 0.8982 |

Table 6: Transfer learning results for objects detection task (pretrained on ImageNet). Numbers are mAP for VOC2007; mean-per-class accuracy for Aircraft and Pets; and top-1 accuracy for Birdsnap.

highest Top-1 accuracy of 0.743 with only 10% of labels and maintains a strong lead at 0.760 with 50% labels, while also significantly improving Top-5 performance. On ImageNet, CLOP achieved a Top-1 accuracy of 0.791 at 10% label availability and 0.829 at 50%, outperforming the next best methods by margins of over 4% in some cases.

For imbalanced-class training, we generate class-wise sample ratios by sampling from a uniform distribution in the range of $[0.001, 1]$, while maintaining all other experimental settings identical to the balanced-class training setup. The results are presented in Table 4, demonstrating that CLOP significantly outperforms all benchmark methods.

TRANSFER LEARNING ON IMAGE CLASSIFICATION AND OBJECT DETECTION

To assess CLOP's generalization capability on unseen datasets, we conduct transfer learning experiments on both image classification and object detection tasks. Specifically, we first pretrain the ResNet-50 (4x) backbone using various contrastive loss functions on ImageNet with all labels. Subsequently, we replace the projection head with either a one-layer prediction head or a two-layer object detection head with ReLU activation and fine-tune the network on the nature image datasets, including Food Bossard et al. (2014), CIFAR-10 Krizhevsky et al. (2010), CIFAR-100 Krizhevsky et al. (2009), SUN397 Xiao et al. (2010), DTD Qu et al. (2023), Caltech-101 Fei-Fei et al. (2004), Flowers Nilsback & Zisserman (2008), Birdsnap Berg et al. (2014), Cars Yang et al. (2015), Aircraft Maji et al. (2013), VOC2007 Everingham (2007), Pets Patino et al. (2016). For image classification tasks, we report the accuracy results in Table 5. Following the standard evaluation metrics in Chen et al. (2020); Khosla et al. (2020), we present mean-per-class accuracy for Caltech and Flowers, while reporting top-1 accuracy for all other datasets. The results indicate that CLOP generally outperforms competing methods. Although SimMatch and SimMatch-V2 achieve slightly higher accuracy on the Food and DTD datasets, the performance gain is less than 2%. In contrast, CLOP surpasses these methods on the remaining four datasets, with the most significant improvement exceeding 3.5% on CIFAR-100. For object detection tasks, we report mean average precision (mAP) for VOC2007, mean-per-class accuracy for Aircraft and Pets, and top-1 accuracy for Birdsnap. The results, presented in Table 6, demonstrate that CLOP outperforms competing methods on three out of five datasets.

| Statistic | Non-Orthonormal | Orthonormal |
|---|---|---|
| Mean | 0.768 | **0.780** |
| Median | 0.762 | **0.781** |
| Std | 0.015 | **0.009** |
| Lower Quantile | 0.760 | **0.775** |
| Upper Quantile | 0.775 | **0.786** |

Table 7: Ablation study on the effect of prototype initialization. We compare non-orthonormal and orthonormal initialization over 10 independent runs on CIFAR-100 under the full label setting. Results report the distribution statistics of Top-1 classification accuracy.

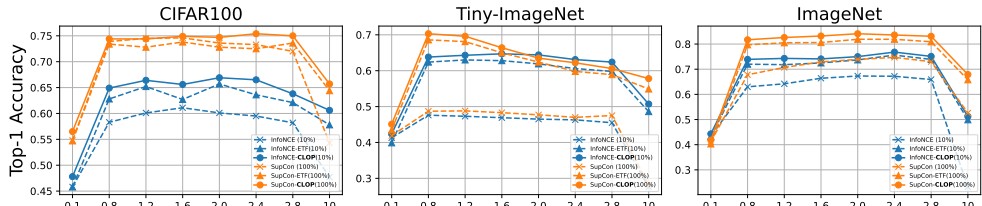

Figure 2: Top-1 classification accuracy across different learning rates. The percentage of labels used for supervised training is indicated in the legend.

## ABLATION STUDIES

In this section, we present the experimental results for image classification, conducted with various batch sizes and learning rates on the CIFAR-100 and ImageNet datasets. For baseline methods, we implement the InfoNCE Wu et al. (2018) with a supervised linear classifier for semi-supervised learning and the SupCon Khosla et al. (2020) for fully-supervised learning. All experiments are performed using the SimCLR Chen et al. (2020) framework with ResNet-50 (4x) He et al. (2016).

**Effect of Prototype Initialization.** To assess the impact of prototype initialization in our framework, we conduct an ablation study comparing non-orthonormal prototypes with orthonormal prototypes on CIFAR-100 under full supervision. As shown in Table 7, initializing prototypes with orthonormal vectors consistently yields better performance across all statistical measures. Specifically, orthonormal initialization improves the mean top-1 accuracy from 0.768 to 0.780 and reduces the standard deviation from 0.015 to 0.009, indicating both improved performance and training stability.

**Orthonormal (CLOP) vs ETF prototypes.** To ensure a fair comparison with ETF, we also evaluate performance using ETF prototypes as an alternative to the orthonormal prototypes. For fully-supervised learning, we utilize all labels in the training datasets for both SupCon and CLOP. In the semi-supervised setting, we employ 10% of the labeled data for both linear classifier and CLOP training. We report top-1 classification accuracies on ImageNet, CIFAR100 and Tiny-ImageNet in Figure 2 and top-5 accuracies in Appendix using the supervised linear classifier.

**CLOP Prevents Collapse with Large Learning Rates.** We trained models with learning rates ranging from 0.1 to 10 on CIFAR-100 and ImageNet for 200 epochs and Tiny-ImageNet for 100 epochs, using a batch size of 1024. The corresponding classification accuracies are presented in Figure 2 for top-1 and in Appendix for top-5 accuracies. Across both datasets, CLOP consistently outperforms the baseline methods. Moreover, as demonstrated by Section , excessively large learning

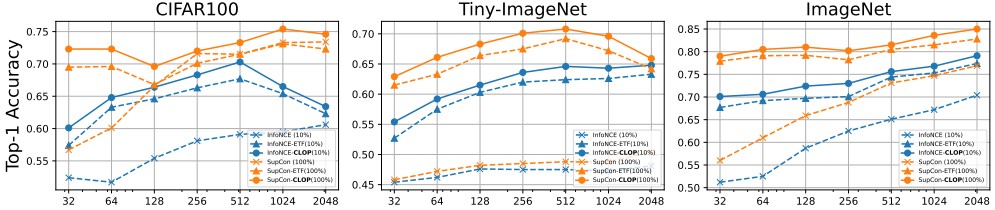

Figure 3: Top-1 classification accuracy across different batch sizes. The percentage of labels used for supervised training is indicated in the legend.

Table 8: Ablation studies on $\lambda$, similarity metric for CLOP, and augmentation strategies.

(a) Acc of different $\lambda$.

| $\lambda$ | Top-1 | Top-5 |
|---|---|---|
| 0.1 | 0.745 | 0.935 |
| 0.5 | 0.740 | 0.931 |
| 1.0 | 0.754 | **0.938** |
| 1.5 | **0.760** | 0.937 |

(b) Acc of different similarity metric.

| Similarity Metric | Top-1 | Top-5 |
|---|---|---|
| Cosine | **0.754** | **0.938** |
| Euclidean | 0.749 | 0.933 |
| Manhattan | 0.715 | 0.899 |

(c) Acc of augmentation strategies.

| Augmentation | Top-1 | Top-5 |
|---|---|---|
| AutoAugment | 0.625 | 0.847 |
| SimCLR | **0.754** | **0.938** |
| RandAugment | 0.726 | 0.899 |

rates can lead to complete collapse, as clearly observed in the baseline methods at a learning rate of 10 on both datasets. However, with the incorporation of CLOP into the loss function, we observe a significantly smaller performance degradation on both datasets.

**CLOP Enables Smaller Batch Sizes.** We trained models with batch sizes of 32, 64, 128, 256, 512, 1024, and 2048 on CIFAR-100 and ImageNet for 200 epochs and on Tiny-ImageNet for 100 epochs. The learning rate was fixed at $(0.3 \times \text{batch size}/256)$ for optimal performance. The corresponding top-1 classification accuracies are presented in Figure 3 and top-5 accuracies are in Appendix. CLOP consistently outperformed the baseline methods across all batch sizes. As reported in the original papers Chen et al. (2020); Khosla et al. (2020), contrastive learning performs optimally when the batch size exceeds 1024, a finding corroborated by our experiments. However, with the addition of CLOP, we observe significantly less performance degradation at smaller batch sizes. Remarkably, CLOP achieved similar accuracy with a batch size of 32 compared to the baseline SupCon with a batch size of 2048 for CIFAR-100.

**Tuning $\lambda$.** To evaluate the sensitivity of the tuning parameter $\lambda$ in CLOP, we trained the model with SupCon loss across different $\lambda$ values, keeping the batch size fixed at 1024. The classification accuracy on both CIFAR-100 is reported in Table 8(a). We observe that the performance remains stable for $\lambda$ values ranging from 0.1 to 1.5, with $\lambda = 1.0$ and $\lambda = 1.5$ yielding the best overall performance.

**Choice of Similarity Metric.** To evaluate the impact of different similarity functions on Eq. (2), we trained the same ResNet-50 (4x) architecture on CIFAR-100 using cosine similarity, Euclidean similarity, and Manhattan similarity. The results, presented in Table 8(b), indicate that cosine similarity, which aligns with $\mathcal{L}_{CL}$ in Eq. (2), achieves the highest performance.

**Augmentation Strategies.** To evaluate the impact of augmentation strategies on CLOP, we trained the same ResNet-50 (4x) model on CIFAR-100 with a batch size of 1024. We selected three commonly used augmentation methods: 1) RandAugment: Augmentation with three operations randomly chosen from all image processing functions in PyTorch (e.g., padding, resizing, cropping, rotation, color jitter, Gaussian blur, inversion, contrast adjustment, equalization); 2) AutoAugment using the ImageNet policy proposed in Cubuk et al. (2018); 3) SimCLR Augmentation Policy from Chen et al. (2020). The results are shown in Table 8(c), indicate that SimCLR augmentation works best with CLOP.

## CONCLUSION AND LIMITATIONS

In this work, we addressed the challenge of dimensional collapse in contrastive learning losses based solely on cosine similarity. We first demonstrated that such losses admit degenerate stationary points, where embeddings collapse into a rank-1 subspace and yield uninformative representations. To mitigate this issue, we introduced CLOP, a novel semi-supervised contrastive loss that aligns embeddings with orthonormal prototypes, thereby preserving representational diversity. Extensive experiments on image classification and object detection, under both balanced and imbalanced label regimes, confirmed that CLOP consistently outperforms strong baselines. Beyond higher accuracy, CLOP demonstrates robustness to reduced batch sizes and large learning rates, making it practical for real-world training scenarios. While CLOP assumes a fixed number of well-separated classes and relies on static prototype initialization, our results highlight its effectiveness as a step toward more stable and generalizable contrastive learning. Future directions include adaptive prototype mechanisms and extensions to fine-grained, hierarchical, or evolving label structures.

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

## PROOF OF LEMMA 1

*Proof of Lemma 1.* Consider $\mathcal{L}_i$ as the i-th loss term of $\mathcal{L}_{\text{InfoNCE}}$, defined by the following expression:

$$\mathcal{L}_i := -\log \mathbb{P}_i$$

where $\mathbb{P}_i$ denotes the probability that i-th embedding choose its positive pair as closest neighbor:

$$\mathbb{P}_i := \frac{\exp(\mathbf{z}_i^\top \mathbf{z}_{j(i)}/\tau)}{\exp(\mathbf{z}_i^\top \mathbf{z}_{j(i)}/\tau) + \sum_{a \notin \{i,j(i)\}} \exp(\mathbf{z}_i^\top \mathbf{z}_a/\tau)}$$

As detailed in Yeh et al. (2022), the gradient of $\mathcal{L}_i$ with respect to $\mathbf{z}_i$, $\mathbf{z}_{j(i)}$, and $\mathbf{z}_a$ can be derived as follows:

$$-\frac{\partial \mathcal{L}_i}{\partial \mathbf{z}_i} := (1 - \mathbb{P}_i)/\tau \left( \mathbf{z}_{j(i)} - \sum_{a \notin \{i,j(i)\}} \frac{\exp(\mathbf{z}_i^\top \mathbf{z}_a/\tau)}{\sum_{b \notin \{i,j(i)\}} \exp(\mathbf{z}_i^\top \mathbf{z}_b/\tau)} \mathbf{z}_a \right)$$

$$-\frac{\partial \mathcal{L}_i}{\partial \mathbf{z}_{j(i)}} := \frac{(1 - \mathbb{P}_i)}{\tau} \mathbf{z}_i$$

$$-\frac{\partial \mathcal{L}_i}{\partial \mathbf{z}_a} := -\frac{(1 - \mathbb{P}_i)}{\tau} \frac{\exp(\mathbf{z}_i^\top \mathbf{z}_a/\tau)}{\sum_{b \notin \{i,j(i)\}} \exp(\mathbf{z}_i^\top \mathbf{z}_b/\tau)} \mathbf{z}_i$$

In the standard setup of self-supervised learning, for any sample, there is one positive pair among $I$ and the remainder are all negative pairs. By aggregating all the gradient respect to a single sample, we have the gradient of InfoNCE respect to $\mathbf{z}_i$:

$$-\frac{\partial \mathcal{L}_{\text{InfoNCE}}}{\partial \mathbf{z}_i} := \frac{(1 - \mathbb{P}_i) + (1 - \mathbb{P}_{j(i)})}{\tau} \mathbf{z}_{j(i)} - \sum_{a \notin \{i,j(i)\}} \frac{(1 - \mathbb{P}_i)}{\tau} \frac{\exp(\mathbf{z}_i^\top \mathbf{z}_a/\tau)}{\sum_{b \notin \{i,j(i)\}} \exp(\mathbf{z}_i^\top \mathbf{z}_b/\tau)} \mathbf{z}_a$$

$$- \sum_{a \notin \{i,j(i)\}} \frac{(1 - \mathbb{P}_a)}{\tau} \frac{\exp(\mathbf{z}_i^\top \mathbf{z}_a/\tau)}{\sum_{b \notin \{a,j(a)\}} \exp(\mathbf{z}_a^\top \mathbf{z}_b/\tau)} \mathbf{z}_a$$

Now, considering the first scenario, where all embeddings equal, that means that $\mathbf{z}_i = \mathbf{z}_{j(i)} = \mathbf{z}_a = \mathbf{z}^*$ for all $a \in I$, the loss terms $\mathbb{P}_i$, $\mathbb{P}_{j(i)}$, and $\mathbb{P}_a$ converge to a constant $\mathbb{P}^*$, given by:

$$\mathbb{P}_i = \mathbb{P}_{j(i)} = \mathbb{P}_a = -\log \frac{1}{|I| - 1} := \mathbb{P}^*$$

Consequently, the gradient of $\mathcal{L}_{\text{InfoNCE}}$ with respect to $\mathbf{z}_i$ under this assumption reduces to zero, aligning with our expectations:

$$-\frac{\partial \mathcal{L}_{\text{InfoNCE}}}{\partial \mathbf{z}_i} = \frac{2(1 - \mathbb{P}^*)}{\tau} \mathbf{z}^* - 2(|I| - 2) \frac{(1 - \mathbb{P}^*)}{\tau} \frac{1}{|I| - 2} \mathbf{z}^* = 0$$

We establish the existence of local minima in scenarios where all embeddings are identical. Now, we consider the second scenario where all embeddings generated reside within the same rank-1 subspace. Denoting $\mathbf{z}^*$ as their unit basis, we can represent each embedding $\mathbf{z}_i$ as:

$$\mathbf{z}_i = \alpha \mathbf{z}^*, \quad \alpha \in \{-1, 1\}, \quad \forall i$$

The gradient of the loss function $\mathcal{L}_{\text{InfoNCE}}$ with respect to $\mathbf{z}_i$ simplifies to:

$$-\frac{\partial \mathcal{L}}{\partial \mathbf{z}_i} = \beta \mathbf{z}_i$$

Here, $\beta$ is a scalar that aggregates contributions from all relevant weights.

It is important to note that $\mathbf{z}^i$ represents the normalized output of the function $f$, with $\mathbf{x}_i$ denoting the original, unnormalized embedding. This implies the following relation:

$$-\frac{\partial \mathcal{L}}{\partial \mathbf{x}_i} = -\frac{\partial \mathcal{L}}{\partial \mathbf{z}_i} \frac{\partial \mathbf{z}_i}{\partial \mathbf{x}_i} = \frac{1}{\|\mathbf{x}_i\|_2} \left( \mathbb{I} - \frac{\mathbf{x}_i \mathbf{x}_i^\top}{\|\mathbf{x}_i\|_2^2} \right) \beta \mathbf{z}_i = \frac{\beta}{\|\mathbf{x}_i\|_2} \left( \mathbf{z}_i - \frac{\mathbf{x}_i}{\|\mathbf{x}_i\|_2} \right) = 0,$$

where $\mathbb{I}$ represents the identity matrix.

$\square$

## GRADIENT ANALYSIS OF REPULSIVE FORCE FROM INFONCE LOSS

Recall that

$$\mathcal{L}_{\text{infoNCE}} = -\sum_{i \in I} \log \frac{\exp(\mathbf{z}_i^\top \mathbf{z}_{j(i)}/\tau)}{\sum_{a \neq i} \exp(\mathbf{z}_i^\top \mathbf{z}_a/\tau)}$$

Assume that the model can successfully merge the positive pair into the same embedding, the loss of repulsive becomes:

$$\mathcal{L}_{\text{repulsive}} = -\sum_{i \in I} \log \frac{\exp(1/\tau)}{\sum_{a \neq i,j(i)} \exp(\mathbf{z}_i^\top \mathbf{z}_a/\tau) + \exp(1/\tau)}$$

We start by rewriting the loss for a given sample i:

$$\ell_i = -\log \frac{\exp(1/\tau)}{\exp(1/\tau) + \sum_{a \in \mathcal{N}(i)} \exp\left(\mathbf{z}_i^\top \mathbf{z}_a/\tau\right)},$$

where we denote

$$\mathcal{N}(i) := \{\, a : a \neq i \text{ and } a \neq j(i) \,\}.$$

Define the denominator

$$D_i := \exp(1/\tau) + \sum_{a \in \mathcal{N}(i)} \exp\left(\mathbf{z}_i^\top \mathbf{z}_a/\tau\right).$$

Then

$$\ell_i = \log\left[D_i\right] - \frac{1}{\tau}.$$

Define $\mathbb{P}_{ia}$ as the probability that sample $i$ choose sample $a$ as closest neighbor,

$$\mathbb{P}_{ia} := \frac{\exp\left(\mathbf{z}_i^\top \mathbf{z}_a/\tau\right)}{\exp(1/\tau) + \sum_{a \in \mathcal{N}(i)} \exp\left(\mathbf{z}_i^\top \mathbf{z}_a/\tau\right)} = \frac{\exp\left(\mathbf{z}_i^\top \mathbf{z}_a/\tau\right)}{D_i}$$

**Taking the gradient of $\ell_i$ with respect to $\mathbf{z}_i$,**

$$\frac{\partial}{\partial \mathbf{z}_i} \exp\left(\mathbf{z}_i^\top \mathbf{z}_a/\tau\right) = \frac{1}{\tau} \exp\left(\mathbf{z}_i^\top \mathbf{z}_a/\tau\right) \mathbf{z}_a.$$

Thus, differentiating the log term gives:

$$\frac{\partial \ell_i}{\partial \mathbf{z}_i} = \frac{1}{D_i} \sum_{a \in \mathcal{N}(i)} \frac{1}{\tau} \exp\left(\mathbf{z}_i^\top \mathbf{z}_a/\tau\right) \mathbf{z}_a.$$

Rewriting in terms of the softmax probabilities,

$$\frac{\partial \ell_i}{\partial \mathbf{z}_i} = \frac{1}{\tau} \sum_{a \in \mathcal{N}(i)} \mathbb{P}_{ia} \mathbf{z}_a,$$

**Taking the gradient of $\ell_i$ with respect to $\mathbf{z}_a$,**

$$\frac{\partial}{\partial \mathbf{z}_a} \exp\left(\mathbf{z}_i^\top \mathbf{z}_a/\tau\right) = \frac{1}{\tau} \exp\left(\mathbf{z}_i^\top \mathbf{z}_a/\tau\right) \mathbf{z}_i.$$

Thus, differentiating the log term gives:

$$\frac{\partial \ell_i}{\partial \mathbf{z}_a} = \frac{1}{D_i} \frac{1}{\tau} \exp\left(\mathbf{z}_i^\top \mathbf{z}_a/\tau\right) \mathbf{z}_i.$$

Rewriting in terms of the softmax probabilities,

$$\frac{\partial \ell_i}{\partial \mathbf{z}_a} = \frac{1}{\tau} \mathbb{P}_{ia} \mathbf{z}_i.$$

Therefore, the gradient of $\mathcal{L}_{\text{repulsive}}$ is:

$$\frac{\partial \mathcal{L}_{\text{repulsive}}}{\partial \mathbf{z}_i} = \frac{\partial \ell_i}{\partial \mathbf{z}_i} + \sum_{a \in \mathcal{N}(i)} \frac{\partial \ell_a}{\partial \mathbf{z}_i} = \frac{1}{\tau} \sum_{a \in \mathcal{N}(i)} \mathbb{P}_{ia} \, \mathbf{z}_a + \frac{1}{\tau} \sum_{a \in \mathcal{N}(i)} \mathbb{P}_{ai} \, \mathbf{z}_a = \frac{1}{\tau} \sum_{a \in \mathcal{N}(i)} \left( \mathbb{P}_{ia} + \mathbb{P}_{ai} \right) \mathbf{z}_a$$

Since the probability matrix $\mathbb{P}$ is obviously symmetric,

$$\frac{\partial \mathcal{L}_{\text{repulsive}}}{\partial \mathbf{z}_i} = \frac{2}{\tau} \sum_{a \in \mathcal{N}(i)} \mathbb{P}_{ia} \mathbf{z}_a$$

Thus, for the gradient of the before the normalization,

$$\frac{\partial \mathcal{L}}{\partial \mathbf{x}_i} = \frac{\partial \mathcal{L}}{\partial \mathbf{z}_i} \frac{\partial \mathbf{z}_i}{\partial \mathbf{x}_i} = \frac{1}{\|\mathbf{x}_i\|_2} \left( \mathbb{I} - \frac{\mathbf{x}_i (\mathbf{x}_i)^\top}{\|\mathbf{x}_i\|_2^2} \right) \frac{2}{\tau} \sum_{a \in \mathcal{N}(i)} \mathbb{P}_{ia} \mathbf{z}_a = \frac{1}{\|\mathbf{x}_i\|_2} \left( \mathbb{I} - \mathbf{z}_i (\mathbf{z}_i)^\top \right) \frac{2}{\tau} \sum_{a \in \mathcal{N}(i)} \mathbb{P}_{ia} \mathbf{z}_a$$

Given the first iteration of points as $\{\mathbf{x}_i^0\}$, then, after the first iteration of descent, the next embeddings are:

$$\mathbf{x}_i^1 = \mathbf{x}_i^0 - \eta \frac{1}{\|\mathbf{x}_i^0\|_2} \left( \mathbb{I} - \mathbf{z}_i^0 (\mathbf{z}_i^0)^\top \right) \frac{2}{\tau} \sum_{a \in \mathcal{N}(i)} \mathbb{P}_{ia} \, \mathbf{z}_a^0$$

Thus, the mean of $\mathbf{x}_i^1$ can be calculated as:

$$\boldsymbol{\mu}^1 = \frac{1}{N} \sum_{i=1}^N \mathbf{x}_i^1 = \frac{1}{N} \sum_{i=1}^N \mathbf{x}_i^0 - \frac{2\eta}{N\tau} \sum_{i=1}^N \sum_{a \in \mathcal{N}(i)} \frac{1}{\|\mathbf{x}_i^0\|_2} \mathbb{P}_{ia} \left( \mathbb{I} - \mathbf{z}_i^0 (\mathbf{z}_i^0)^\top \right) \mathbf{z}_a^0$$

Since the negative pair relationship is symmetric, ie. $a \in \mathcal{N}(i) \iff i \in \mathcal{N}(a)$, then

$$\sum_{i=1}^N \sum_{a \in \mathcal{N}(i)} \mathbb{P}_{ia} \frac{1}{\|\mathbf{x}_i^0\|_2} \left( \mathbb{I} - \mathbf{z}_i^0 (\mathbf{z}_i^0)^\top \right) \mathbf{z}_a^0 = \sum_{a=1}^N \sum_{i \in \mathcal{N}(a)} \mathbb{P}_{ia} \frac{1}{\|\mathbf{x}_i^0\|_2} \left( \mathbb{I} - \mathbf{z}_i^0 (\mathbf{z}_i^0)^\top \right) \mathbf{z}_a^0$$

Thus,

$$
\boldsymbol{\mu}^1 = \frac{1}{N} \sum_{i=1}^{N} \mathbf{x}_i^0 - \frac{2\eta}{N\tau} \sum_{a=1}^{N} \sum_{i \in \mathcal{N}(a)} \mathbb{P}_{ia} \frac{1}{\|\mathbf{x}_i^0\|_2} \left( \mathbb{I} - \mathbf{z}_i^0 (\mathbf{z}_i^0)^\top \right) \mathbf{z}_a^0
$$

$$
= \frac{1}{N} \sum_{i=1}^{N} \mathbf{x}_i^0 - \frac{2\eta}{N\tau} \sum_{i=1}^{N} \sum_{a \in \mathcal{N}(i)} \frac{\mathbb{P}_{ia}}{\|\mathbf{x}_a^0\|_2} \left( \mathbb{I} - \mathbf{z}_a^0 (\mathbf{z}_a^0)^\top \right) \mathbf{z}_i^0
$$

$$
= \frac{1}{N} \sum_{i=1}^{N} \mathbf{x}_i^0 - \frac{2\eta}{N\tau} \sum_{i=1}^{N} \sum_{a \in \mathcal{N}(i)} \frac{\mathbb{P}_{ia}}{\|\mathbf{x}_a^0\|_2} \mathbf{z}_i^0 + \frac{2\eta}{N\tau} \sum_{i=1}^{N} \sum_{a \in \mathcal{N}(i)} \frac{\mathbb{P}_{ia}}{\|\mathbf{x}_a^0\|_2} \mathbf{z}_a^0 (\mathbf{z}_a^0)^\top \mathbf{z}_i^0
$$

$$
= \frac{1}{N} \sum_{i=1}^{N} \mathbf{x}_i^0 - \frac{2\eta}{N\tau} \sum_{i=1}^{N} \sum_{a \in \mathcal{N}(i)} \frac{\mathbb{P}_{ia}}{\|\mathbf{x}_a^0\|_2} \mathbf{z}_i^0 + \frac{2\eta}{N\tau} \sum_{i=1}^{N} \sum_{a \in \mathcal{N}(i)} \frac{\mathbb{P}_{ia}}{\|\mathbf{x}_i^0\|_2} \mathbf{z}_i^0 (\mathbf{z}_i^0)^\top \mathbf{z}_a^0
$$

$$
= \frac{1}{N} \sum_{i=1}^{N} \mathbf{x}_i^0 - \frac{2\eta}{N\tau} \sum_{i=1}^{N} \sum_{a \in \mathcal{N}(i)} \mathbb{P}_{ia} \frac{1}{\|\mathbf{x}_a^0\|_2} \mathbf{z}_i^0 + \frac{2\eta}{N\tau} \sum_{i=1}^{N} \sum_{a \in \mathcal{N}(i)} \mathbb{P}_{ia} \frac{(\mathbf{z}_i^0)^\top \mathbf{z}_a^0}{\|\mathbf{x}_i^0\|_2} \mathbf{z}_i^0
$$

$$
= \frac{1}{N} \sum_{i=1}^{N} \mathbf{x}_i^0 - \frac{2\eta}{N\tau} \sum_{i=1}^{N} \sum_{a \in \mathcal{N}(i)} \mathbb{P}_{ia} \left( \frac{1}{\|\mathbf{x}_a^0\|_2} - \frac{(\mathbf{z}_i^0)^\top \mathbf{z}_a^0}{\|\mathbf{x}_i^0\|_2} \right) \frac{1}{\|\mathbf{x}_i^0\|_2} \mathbf{x}_i^0
$$

$$
= \frac{1}{N} \sum_{i=1}^{N} \left[ \left( 1 - \frac{2\eta}{\tau} \right) \frac{1}{\|\mathbf{x}_i^0\|_2} \sum_{a \in \mathcal{N}(i)} \mathbb{P}_{ia} \left( \frac{1}{\|\mathbf{x}_a^0\|_2} - \frac{(\mathbf{z}_i^0)^\top \mathbf{z}_a^0}{\|\mathbf{x}_i^0\|_2} \right) \right] \mathbf{x}_i^0
$$

$$
= \left( 1 - \frac{2\eta}{\tau} \right) \frac{1}{N} \sum_{i=1}^{N} \left[ \frac{1}{\|\mathbf{x}_i^0\|_2} \sum_{a \in \mathcal{N}(i)} \mathbb{P}_{ia} \left( \frac{1}{\|\mathbf{x}_a^0\|_2} - \frac{(\mathbf{z}_i^0)^\top \mathbf{z}_a^0}{\|\mathbf{x}_i^0\|_2} \right) \right] \mathbf{x}_i^0
$$

Since $-1 \le \mathbf{z}_i^\top \mathbf{z}_a \le 1$, Denote

$$
\beta_{ia} := \left[ \frac{1}{\|\mathbf{x}_i^0\|_2} \sum_{a \in \mathcal{N}(i)} \mathbb{P}_{ia} \left( \frac{1}{\|\mathbf{x}_a^0\|_2} - \frac{(\mathbf{z}_i^0)^\top \mathbf{z}_a^0}{\|\mathbf{x}_i^0\|_2} \right) \right]
$$

$$
\le \frac{1}{\|\mathbf{x}_i^0\|_2} \sum_{a \in \mathcal{N}(i)} \mathbb{P}_{ia} \left( \frac{1}{\|\mathbf{x}_a^0\|_2} + \frac{1}{\|\mathbf{x}_i^0\|_2} \right)
$$

$$
\le \frac{2}{\min_j \|\mathbf{x}_j^0\|_2^2} \sum_{a \in \mathcal{N}(i)} \mathbb{P}_{ia}
$$

And we can lowerbound sum of probability by

$$
\sum_{a \in \mathcal{N}(i)} \mathbb{P}_{ia} = \frac{\sum_{a \in \mathcal{N}(i)} \exp\left( (\mathbf{z}_i^0)^\top (\mathbf{z}_a^0)/\tau \right)}{\exp(1/\tau) + \sum_{a \in \mathcal{N}(i)} \exp\left( (\mathbf{z}_i^0)^\top (\mathbf{z}_a^0)/\tau \right)} \le \frac{|\mathcal{N}| \exp(1/\tau)}{\exp(1/\tau) + |\mathcal{N}| \exp(1/\tau)} = \frac{|\mathcal{N}|}{1 + |\mathcal{N}|}
$$

Thus, denote $\sigma := \min_j \|\mathbf{x}_j^0\|_2$, we show that

$$
\beta_{ia} \le \frac{2}{\sigma^2} \frac{|\mathcal{N}|}{1 + |\mathcal{N}|}
$$

Therefore,

$$
\|\boldsymbol{\mu}_1\|_2 \le \left| \left( 1 - \frac{2\eta}{\tau} \right) \frac{2}{\sigma^2} \frac{|\mathcal{N}|}{1 + |\mathcal{N}|} \right| \|\boldsymbol{\mu}_0\|_2
$$

If we want to limit the increase of mean, we want the coefficient to be less than 1, which means that

$$-\frac{\sigma^2(1+|\mathcal{N}|)}{2|\mathcal{N}|} < 1 - \frac{2\eta}{\tau} < \frac{\sigma^2(1+|\mathcal{N}|)}{2|\mathcal{N}|}$$

$$-1 - \frac{\sigma^2(1+|\mathcal{N}|)}{2|\mathcal{N}|} < -\frac{2\eta}{\tau} < -1 + \frac{\sigma^2(1+|\mathcal{N}|)}{2|\mathcal{N}|}$$

$$1 + \frac{\sigma^2(1+|\mathcal{N}|)}{2|\mathcal{N}|} > \frac{2\eta}{\tau} > 1 - \frac{\sigma^2(1+|\mathcal{N}|)}{2|\mathcal{N}|}$$

$$\frac{\tau}{2}\left(1 + \frac{\sigma^2(1+|\mathcal{N}|)}{2|\mathcal{N}|}\right) > \eta > \frac{\tau}{2}\left(1 - \frac{\sigma^2(1+|\mathcal{N}|)}{2|\mathcal{N}|}\right)$$

## EXTRA EXPERIMENTS ON CLOP

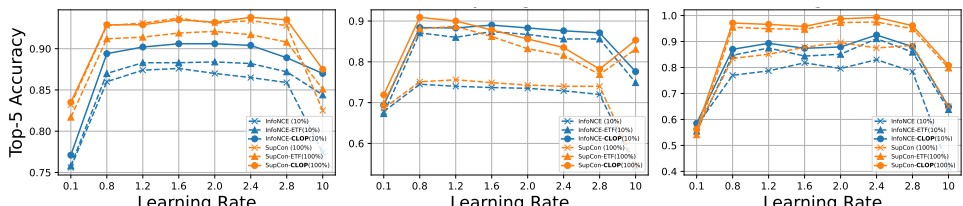

Figure 4: Top-5 classification accuracy across different learning rates. The percentage of labels used for supervised training is indicated in the legend.

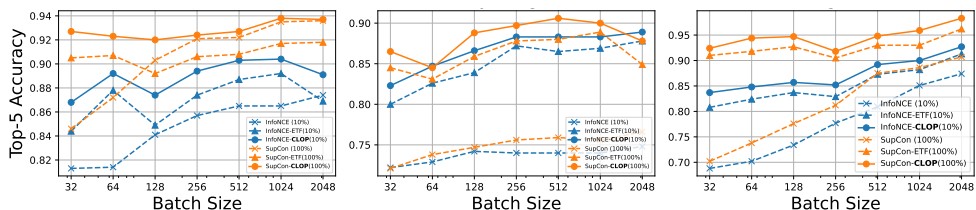

Figure 5: Top-5 classification accuracy across different batch sizes. The percentage of labels used for supervised training is indicated in the legend.

