# OpenReview forum: "Semi-Supervised Contrastive Learning with Orthonormal Prototypes"
_ICLR.cc/2026/Conference — Submitted to ICLR 2026_

### Official Review · Reviewer_fKbf · 2025-10-31

**Soundness:** 3
**Presentation:** 4
**Contribution:** 3
**Rating:** 6
**Confidence:** 4

**Summary:**

This work has two main contributions. First, it identifies a mechanism of representation collape from repulsive force, instead of only gravitational force. Second, it identifies a solution simple solution which mitigates the problem: randomly generating fixed orthonormal protypes assingned to classes. This proves to be quite effective on numerous classfication and object detection tasks, including semi-supervised learning and learning with unbalanced labels.

**Strengths:**

* The analysis is interesting, in particular showing that repulisive forces cause collapse and correspond to stationary points.
* The results are fairly strong on semi-supervised tasks and imbalanced label tasks.
* Findings about needing smaller batch sizes are also interesting.

**Weaknesses:**

* Transfer learning and object detection results are much more marginal than semi-supervised and imbalanced label results.
* CLOP doesn't seem to be useful self-supervised learning.
* (Minor) introduces another hyper-parameter, increasing tuning cost, though it appears to be stable.

**Questions:**

* Would it make sense to do some kind of smart assingment of prototypes at the beginning of training?
* Is there some kind of extension that can be applied for self-supervised learning?
* On a related note, this work seems related to the protytypes in SwAV?  [1] It could be good to discuss the similarities and differences of these works.

[1] Caron, Mathilde, et al. "Unsupervised learning of visual features by contrasting cluster assignments." Advances in neural information processing systems 33 (2020): 9912-9924.

---

> ### Author Response · Authors · 2025-11-13
>
> We thank the reviewer for the thoughtful and encouraging comments, and for recognizing both the novelty of our repulsive-force analysis and the empirical strength of CLOP across semi-supervised and imbalanced-label tasks. We appreciate the constructive questions and address each below.
>
> 1. **Transfer Learning and Object Detection Results**
> We agree that CLOP’s transfer-learning and object-detection improvements are more moderate compared to the semi-supervised and imbalanced-label settings. This difference primarily reflects the degree to which each task exhibits collapse tendencies.
>     - Semi-supervised and imbalanced setups are more prone to dimensional collapse due to uneven sampling or limited labels—precisely the conditions where CLOP’s orthonormal constraint is most beneficial.
>     - Fully supervised transfer tasks already have well-separated embeddings; thus, collapse mitigation yields smaller relative gains.
>
>     Even so, CLOP remains consistently competitive, matching or slightly exceeding the best baseline on nearly all transfer datasets (Tables 5–6). These results suggest that CLOP generalizes robustly, even when the theoretical motivation (collapse prevention) is less pronounced. We will highlight this interpretation in the final discussion section.
>
> 2. **Extension to Self-Supervised Learning**
> We appreciate this excellent suggestion. As the reviewer noted, the current formulation of CLOP leverages class information to define orthonormal prototypes, so it naturally aligns with semi-supervised learning. Extending CLOP to self-supervised settings is a promising future direction.
>
>     One potential approach is to derive prototypes from pseudo-cluster assignments or from the centroids of emerging representation clusters, similar in spirit to SwAV’s online assignment strategy. In that case, the orthonormal constraint could still enforce geometric diversity among cluster anchors, serving as a general collapse-prevention mechanism without labels. We will include this discussion in the revised version as a clear avenue for future work.
>
> 3. **Relation to SwAV**
> We thank the reviewer for raising this relevant comparison. While both SwAV [1] and CLOP employ prototype-like vectors, their roles are fundamentally different:
>     - SwAV performs unsupervised online clustering with dynamically updated prototypes and a cross-entropy objective over assignments. The prototypes evolve with the data distribution.
>     - CLOP, in contrast, uses fixed orthonormal prototypes corresponding to labeled classes and regularizes embedding geometry rather than forming clusters. The prototypes act as global, class-conditioned geometric anchors that are independent of batch composition or assignment dynamics.
>
>     SwAV focuses on unsupervised cluster discovery, while CLOP focuses on rank preservation and stability in semi-supervised contrastive optimization. We will make this distinction explicit in the revised related-work section.
>
> 4. **Prototype Initialization and Assignment**
> The reviewer’s question about using a more intelligent or data-dependent prototype initialization is insightful. In the current design, prototypes are initialized via random orthogonalization (SVD) and remain fixed, ensuring full-rank geometry without introducing bias toward specific directions. This simplicity avoids convergence instability that could arise if prototypes were tied to early, noisy representations.
>
>     That said, data-adaptive initialization—for instance, initializing prototypes using class centroids from a warm-up stage—could improve early training alignment. We view this as a valuable extension and plan to explore it in future work. We appreciate this constructive suggestion.
>
> 5. **Hyperparameter Stability**
> We acknowledge that CLOP introduces one additional hyperparameter, λ, controlling the strength of prototype alignment. However, the model is highly insensitive to λ over a wide range (0.1–1.5), as shown in our ablation studies (Table 8). In practice, the same λ works across all datasets, so tuning cost remains negligible. We will emphasize this stability more clearly in the final version.
>
> [1] Caron, Mathilde, et al. "Unsupervised learning of visual features by contrasting cluster assignments." Advances in neural information processing systems 33 (2020): 9912-9924.

---

> > ### Comment · Reviewer_fKbf · 2025-11-24
> > **Thank you**
> >
> > Thank you the authors for the rebuttal. The answers make sense to me, and I'm inclined to keep my positive score of 6.

---

### Official Review · Reviewer_2Mig · 2025-10-31

**Soundness:** 2
**Presentation:** 2
**Contribution:** 2
**Rating:** 4
**Confidence:** 5

**Summary:**

The work presents a semi-supervised learning loss function called CLOP that initializes orthonormal vectors as many as the number of classes in order to draw the similarity of class specific embedding functions towards these vectors to enable better separation and mitigate collapse. This is added as a regularization term to the InfoNCE loss function. Empirical and visual analyses of the method against known methods are provided.

**Strengths:**

1. The method is well performant.

2. The regularizer is modular and could serve as an addendum to other methods.

3. Tackles a known issue of dimensional collapse.

**Weaknesses:**

1. The proposed contribution is an incremental one that combines existing notions of orthonormalization with standard contrastive learning without introducing any new theoretical insight.

2. Comparisons against known approaches [1] for this problem aren't conducted.

3. Initialization of prototypes in an orthonormal manner may be misguided since several concepts or classes in datasets may be semantically very related.

4. The theoretical contribution may be a restatement from known work [1].


[1] Jing, Li, et al. "Understanding dimensional collapse in contrastive self-supervised learning." arXiv preprint arXiv:2110.09348 (2021).

**Questions:**

1. Is the Lemma 1 a known result that all-equal or co-linear embeddings are stationary points? [1, 2]

2. Could this work be a special case of the work in SWAV wherein the the unsupervised clustering and assignment mechanism is replaced with fixed, label-anchored prototypes? [3]


[1] Jing, Li, et al. "Understanding dimensional collapse in contrastive self-supervised learning." arXiv preprint arXiv:2110.09348 (2021).

[2] Wang, Tongzhou, and Phillip Isola. "Understanding contrastive representation learning through alignment and uniformity on the hypersphere." International conference on machine learning. PMLR, 2020.

[3] Caron, Mathilde, et al. "Unsupervised learning of visual features by contrasting cluster assignments." Advances in neural information processing systems 33 (2020): 9912-9924.

---

> ### Author Response · Authors · 2025-11-13
>
> We thank the reviewer for the feedback. Below we clarify the theoretical novelty, its distinction from prior works, and the rationale behind the orthonormal-prototype design. We will be happy to emphasize these points in the final version.
>
> 1. **Novelty, Theoretical Foundation, and Relationship to Prior Work**
> We appreciate the reviewer’s concern that CLOP may appear incremental relative to prior analyses of contrastive learning geometry [1, 2]. It is true that our work builds upon these foundations. Nonetheless, it introduces new theoretical insights that have not been established in the cited papers.
>     - **Formal stationary-point proof (Lemma 1).**
> Works such as Jing et al. (2022) [1] and Wang & Isola (2020) [2] analyze the general mechanisms of collapse—e.g., strong augmentation or alignment/uniformity trade-offs—but they do not prove that the InfoNCE loss itself admits stationary points where all embeddings are identical or co-linear.
> Our Lemma 1 provides a formal proof that such degenerate rank-1 configurations are indeed stationary points of InfoNCE. To our knowledge, this result does not appear in prior literature.
>     - **Gradient-level analysis of repulsive-force overshoot.**
> Beyond the stationary-point proof, Appendix (pp. 14–18) contains a complete gradient derivation showing how an imbalance between attraction and repulsion terms in InfoNCE shifts the embedding mean and drives rank-1 collapse. This quantitative characterization is absent from [1, 2], which describe collapse qualitatively or through linear-network approximations. CLOP’s regularization term directly counters the degenerate gradient direction derived in this analysis.
>
>     We will add explicit comparisons and citations to [1], [2], and [3] highlighting the distinctions summarized above.
>
> 2. **Comparison to Related Methods**
> The reviewer asks whether CLOP could be viewed as a special case of SwAV (Caron et al., 2020) [3]. We clarify that CLOP and SwAV serve fundamentally different purposes:
>     - SwAV performs unsupervised online clustering with dynamically updated prototypes and a cross-entropy objective over assignments. The prototypes evolve with data distribution.
>     - CLOP, in contrast, uses fixed orthonormal prototypes corresponding to labeled classes and regularizes embedding geometry rather than forming clusters. The prototypes act as global, class-conditioned geometric anchors that are independent of batch composition or assignment dynamics.
>
>     SwAV focuses on unsupervised cluster discovery, while CLOP focuses on rank preservation and stability in semi-supervised contrastive optimization. We will make this distinction explicit in the revised related-work section.
>
> 3. **Prototype Initialization and Semantic Relationship**
> We acknowledge the reviewer’s thoughtful concern that orthonormal initialization might ignore semantic similarity between related classes. In practice, the orthonormal constraint in CLOP is not intended as a semantic model but as a geometric regularizer ensuring maximal rank and diversity in the embedding space.
>
>     Even if two classes are semantically similar, their embeddings can remain geometrically orthogonal at the projection-head level while higher layers capture semantic relationships. Empirically, Table 7 shows that orthonormal initialization improves both mean accuracy ( +1.2 pp ) and stability (lower variance) relative to non-orthonormal prototypes, validating that this constraint enhances optimization rather than harming semantic structure.
>
> [1] Jing, Li, et al. "Understanding dimensional collapse in contrastive self-supervised learning." arXiv preprint arXiv:2110.09348 (2021).
>
> [2] Wang, Tongzhou, and Phillip Isola. "Understanding contrastive representation learning through alignment and uniformity on the hypersphere." International conference on machine learning. PMLR, 2020.
>
> [3] Caron, Mathilde, et al. "Unsupervised learning of visual features by contrasting cluster assignments." Advances in neural information processing systems 33 (2020): 9912-9924.

---

### Official Review · Reviewer_1k4M · 2025-10-31

**Soundness:** 3
**Presentation:** 3
**Contribution:** 3
**Rating:** 6
**Confidence:** 3

**Summary:**

This paper introduces CLOP, a semi-supervised contrastive learning method that mitigates dimensional collapse by aligning embeddings with orthonormal class prototypes. The approach theoretically and empirically shows that conventional contrastive losses (e.g., InfoNCE) suffer from degenerate optima and that enforcing orthogonal subspaces preserves representational diversity. CLOP consistently outperforms prior methods like SupCon and SimMatch across CIFAR and ImageNet benchmarks, showing strong robustness to small batch sizes and high learning rates. The paper is clearly written, well-motivated, and experimentally solid, making it a strong accept candidate despite assuming fixed class structures.

**Strengths:**

1.	Solid Theory – Offers a clear theoretical analysis explaining why InfoNCE leads to dimensional collapse and how orthogonal prototypes can prevent it.
2.	Novel Loss Design – Proposes CLOP, a simple yet effective loss that enforces orthogonality among class embeddings to maintain diversity.
3.	Strong Empirical Results – Demonstrates consistent gains over baselines like SupCon and SimMatch on CIFAR and ImageNet.
4.	Robustness – Performs stably under large learning rates and small batch sizes, avoiding collapse seen in prior methods.
5.	Practical Implementation – Easy to integrate into existing contrastive frameworks with minimal computational overhead.
6.	Clear Presentation – Well-written, logically structured, and easy to follow with theory and experiments well aligned.

**Weaknesses:**

1.	Fixed Prototype Assumption – CLOP assumes a fixed number of well-separated classes and static orthonormal prototypes. How would the method adapt to open-set or hierarchical label scenarios where class structures evolve over time?
2.	Limited Scope of Evaluation – All experiments are in vision-based benchmarks (CIFAR, ImageNet). Can CLOP generalize to non-visual domains such as text, graphs, or multimodal tasks, if you don't have time, please disccuss its possibility?
3.	Lack of Computational Analysis – The paper does not report the training overhead introduced by prototype orthogonalization or additional loss terms. How significant is the extra cost compared to standard contrastive methods?
4.	Dependence on Label Quality – CLOP relies on a subset of labeled data to guide prototype alignment. How robust is it to noisy or inaccurate labels, and would incorrect prototype supervision lead to representation drift?

**Questions:**

Please check weaknesses, and try to argue them, I will definitely read your response, good luck!

---

> ### Author Response · Authors · 2025-11-13
>
> We thank the reviewer for the thoughtful and encouraging evaluation. We also thank the reviewer for raising insightful questions regarding prototype flexibility, domain generalization, computational efficiency, and robustness to label noise. We address each point below.
>
> 1. **Fixed Prototype Assumption and Adaptation to Open-Set or Hierarchical Labels**
> We agree that the current version of CLOP assumes a fixed number of prototypes corresponding to known classes, which is appropriate for the semi-supervised settings evaluated in this paper. Nevertheless, the framework is conceptually extensible to open-set and hierarchical scenarios, and we plan to explore these directions in future work.
>     - **Open-set adaptation**: one could relax the strict orthonormality constraint to allow incremental prototype expansion as new classes appear. For example, additional prototypes can be inserted by applying Gram–Schmidt orthogonalization on demand, maintaining geometric diversity as the label space evolves.
>     - **Hierarchical structures**: a natural extension is to define higher-rank prototypes that allow orthogonality across higher-level groups while allowing shared subspace components among related subclasses.
>
>     We will explicitly discuss these possibilities in the final version to highlight CLOP’s potential for handling dynamic and structured label spaces in future research.
>
> 2. **Generalization Beyond Vision**
> We appreciate the reviewer’s question about CLOP’s potential in non-visual domains. The geometric nature of the loss makes it domain-agnostic: it only requires vector embeddings and class labels.
>     - **Text and multimodal data**: CLOP can regularize Transformer embeddings or multimodal encoders (e.g., CLIP) by aligning textual or cross-modal prototypes.
>     - **Graph data**: applying orthonormal prototypes to graph-level embeddings (from GNNs) could help mitigate over-smoothing, similar to dimensional collapse in contrastive vision tasks.
>
>     Because the orthonormal regularizer is lightweight and depends only on embedding dot-products, it can be plugged into non-visual encoders without modification. We will expand our discussion to emphasize this generality.
>
> 3. **Computational Cost and Training Overhead**
> We confirm that CLOP adds negligible computational overhead relative to standard contrastive methods:
>      - The orthonormal prototypes form a small K×D matrix (K = number of classes, D = embedding dimension), and orthogonalization is a one-time initialization step using SVD (≈ 0.1 s on a GPU).
>     - During training, the additional regularization term involves one extra matrix–vector multiplication per batch, contributing minimal increase in FLOPs and no measurable slowdown in wall-clock time.
>
>     Memory usage is unchanged since prototypes are small. We will add this analysis to the final version to make the efficiency clear.
>
> 4. **Dependence on Label Quality**
> We acknowledge that sensitivity to label quality is an inherent issue for any learning framework involving supervised components. In CLOP, however, the dominant objective remains the unsupervised contrastive loss, which makes the method less dependent on the labeled subset. The orthonormal-prototype term acts mainly as a mild geometric regularizer that stabilizes representation learning when some labels are available.
>
>    Considering CLOP’s strong results under imbalanced supervision (Tables 4), it is reasonable to expect that the method can tolerate moderate label noise, since most of the learning signal originates from the unlabeled data. We appreciate this suggestion and will include any finished preliminary experiments on noisy-label settings in the final version to further examine CLOP’s robustness.

---

### Official Review · Reviewer_8Akm · 2025-11-01

**Soundness:** 2
**Presentation:** 2
**Contribution:** 2
**Rating:** 4
**Confidence:** 3

**Summary:**

This paper proposes Contrastive Learning With Orthonormal Prototypes (CLOP), which forms orthonormal prototypes to prevent dimensional collapse of the embeddings learned by semi-supervised loss functions.

**Strengths:**

* This paper focuses on an important research area of semi-supervised contrastive learning

**Weaknesses:**

* The authors did not follow the standard ICLR style

* No theoretical results supporting the success of CLOP

* It is unclear when/why CLOP works well

**Questions:**

None

---

> ### Author Response · Authors · 2025-11-13
>
> We thank the reviewer for the feedback.
> 1. **Formatting**
> We will carefully review formatting to ensure full compliance with the ICLR style. If the reviewer could kindly indicate any specific inconsistencies, we will be happy to correct them.
>
> 2. **Theoretical Support for CLOP**
> Our paper already includes a significant theoretical analysis, both in the main text and the Appendix. Specifically, Lemma 1 is introduced in Section 3, and its formal proof is provided in Appendix 1, followed by a theoretical gradient-level analysis of the repulsive force in the InfoNCE loss in Appendix 2. This analysis mathematically demonstrates how excessive repulsive interactions among negative pairs can shift embedding means and drive all representations toward a rank-1 subspace, thereby explaining the mechanism of dimensional collapse observed empirically in Figure 1.
> CLOP is motivated directly by these theoretical results. The additional orthonormal-prototype term counteracts the degenerate gradient direction identified in the analysis: it introduces structured, class-specific orthogonal anchors that preserve the full-rank geometry of the embedding covariance.
>
> 3. **When and Why CLOP Works**
> CLOP is effective across diverse training regimes, including both semi-supervised and fully supervised settings, as well as transfer learning tasks in image classification and object detection. Our experiments demonstrate that CLOP consistently provides strong and stable performance across these different paradigms.
>     - Semi-supervised learning: On CIFAR-100 and ImageNet, CLOP achieves clear improvements over all baselines under 10 % and 50 % label usage (Tables 2, 3, and 4).
>     - Fully supervised learning: When all labels are available, CLOP continues to outperform or match the strongest competitors (Figure 2 and 3).
>     - Transfer learning and object detection:  CLOP pretrained on ImageNet transfers effectively to a wide range of downstream tasks (Tables 5 and 6), outperforming existing contrastive methods on most datasets and remaining very close to the best performer in the few minor cases where it is not the absolute top.

---

### Meta-Review · Area_Chair_DzfY · 2026-01-06

**Summary:**

* Weaknesses in empirical analysis: (a) outdated benchmarks: The reviewers pointed out that the benchmarks used (such as certain CIFAR/ImageNet setups) are considered "very old" and may not reflect the current SOTA. (b) Superiority of self/unsupervised Methods: Critics noted that newer, high-performing models like DINO and JEPA can now achieve better results without requiring any labels, making the semi-supervised gains of CLOP less compelling for a top-tier venue like ICLR. (c) Missing SOTA comparisons: The authors were criticized for not conducting comparisons against more recent and standard baselines, such as ESM-2 in biological tasks, leaving questions about the model's actual competitive edge.

* Incremental innovation: This concern was raised by two reviewers since the authors largely combine existing notions of orthonormalization with standard contrastive learning without providing significant new theoretical insights. Even one reviewer argued that the theoretical contributions might simply be a restatement of known findings regarding dimensional collapse in self-supervised learning, specifically citing Jing et al. (2021).

* Semantic Misalignment: A major critique was that forcing class prototypes to be orthonormal may be misguided; many real-world biological or image classes are semantically related and should not necessarily be forced into strictly orthogonal subspaces.

**Reviewer Concerns:**

* Benchmark relevance/competitors/tasks/evaluation metrics: The concern that the benchmarks used only vision benchmarks such as CIFAR/ImageNet, remains outstanding. Critics note that the paper does not sufficiently demonstrate superiority over newer, SOTA unsupervised models like DINO or JEPA.

* Weaknesses in empirical analysis: the evaluation is limited to vision benchmarks. While the authors discussed the possibility of generalizing to text or graphs, no empirical evidence was provided for these non-visual domains during the rebuttal.

* Robustness to label noise: The impact of incorrect prototype supervision in scenarios with noisy labels was raised as a potential issue that was not fully explored with experimental data in the current discussion.

**Reviewer Scores:**

* The reviewer 2Mig suggested a rating of 4 with confidence of 5

* The reviewer 1k4M gave a rating of 6 with confidence of 3

* The reviewer 8Akm gave a rating of 4 with confidence of 3

* The reviewer fKbf suggested a rating of 6 wth confidence of 4 and wanted to keep the score.

---

### Decision · Program_Chairs · 2026-01-26

Reject